# Sound Adversarial Audio-Visual Navigation

**Yinfeng Yu[1,3], Wenbing Huang[2], Fuchun Sun[*1], Changan Chen[4], Yikai Wang[1,5], Xiaohong Liu[1]**

[1] Beijing National Research Center for Information Science and Technology (BNRist),
  State Key Lab on Intelligent Technology and Systems,
  Department of Computer Science and Technology, Tsinghua University

[2] Institute for AI Industry Research (AIR), Tsinghua University

[3] College of Information Science and Engineering, Xinjiang University

[4] UT Austin    [5] JD Explore Academy, JD.com

`yyf17@mails.tsinghua.edu.cn, hwenbing@126.com,`
`fcsun@mail.tsinghua.edu.cn`

## Abstract

Audio-visual navigation task requires an agent to find a sound source in a realistic, unmapped 3D environment by utilizing egocentric audio-visual observations. Existing audio-visual navigation works assume a clean environment that solely contains the target sound, which, however, would not be suitable in most real-world applications due to the unexpected sound noise or intentional interference. In this work, we design an acoustically complex environment in which, besides the target sound, there exists a sound attacker playing a zero-sum game with the agent. More specifically, the attacker can move and change the volume and category of the sound to make the agent suffer from finding the sounding object, while the agent tries to dodge the attack and navigate to the goal under the intervention. Under certain constraints to the attacker, we can improve the robustness of the agent towards unexpected sound attacks in audio-visual navigation. For better convergence, we develop a joint training mechanism by employing the property of a centralized critic with decentralized actors. Experiments on two real-world 3D scan datasets (Replica and Matterport3D) verify the effectiveness and the robustness of the agent trained under our designed environment when transferred to the clean environment or the one containing sound attackers with random policy. Project: `https://yyf17.github.io/SAAVN`.

## 1 Introduction

Audio-visual navigation, as currently a vital task for embodied vision (Gordon et al., 2018; Lohmann et al., 2020; Nagarajan & Grauman, 2020), requires the agent to find a sound target in a realistic and unmapped 3D environment by exploring with egocentric audio-visual observations (Chen et al., 2019; Gupta et al., 2017; Chaplot et al., 2020). Inspired by the simultaneous usage of eyes and ears in human exploration (Wilcox et al., 2007; Flom & Bahrick, 2007), audio-visual association benefits the learning of the agent (Gan et al., 2019; 2020a; Dean et al., 2020). A recent work **L**ook, **L**isten, and **A**ct (LLA) has proposed a three-step navigation solution of perception, inference, and decision-making (Gan et al., 2020b). SoundSpaces is the first work to establish an audio-visual embodied navigation simulation platform equipped with the proposed **A**udio-**V**isual embodied **N**avigation (AVN) baseline that resorts to reinforcement learning (Chen et al., 2020). In response to the long-term exploration problem that is caused by the large layout of the 3D scene and the long distance to the target place, **A**udio-**V**isual **Wa**ypoint **N**avigation (AV-WaN) proposes an audio-visual navigation algorithm by setting waypoints as sub-goals to facilitate sound source discovering (Chen et al., 2021b). Besides, **S**emantic **A**udio-**V**isual navi**g**ation (SAVi) develops a navigation algorithm in a scene where the target sound is not periodic and has a variable length; that is, it may stop during the navigation process (Chen et al., 2021a).

Despite the fruitful progress in audio-visual navigation, existing works assume an acoustically simple or clean environment, meaning there is no other sound but the source itself. Nevertheless, this

---
*Corresponding author: Fuchun Sun.

assumption is hardly the case in real life. For example, a kettle in the kitchen beeps to tell the robot that the water is boiling, and the robot in the living room needs to navigate to the kitchen and turn off the stove; while in the living room, two children are playing a game, chuckling loudly from time to time. Such an example poses a crucial challenge on current techniques: can an agent still find its way to the destination without being distracted by all non-target sounds around the agent? Intuitively, the answer is no if the agent has not been trained in acoustically complex environments as in the example listed above. Although the answer is no, this ability is what we expect the agent to possess in real life.

In light of these limitations, we propose first to construct such an acoustically complex environment. In this environment, we add a sound attacker to intervene. This sound attacker can move and change the volume and type of the sound at each time step. In particular, the objective of the sound attacker is to make the agent frustrated by creating a distraction. In contrast, the agent decides how to move at every time step, tries to dodge the sound attack, and explores for the sound target well under the sound attack, as illustrated in Fig. 1. The competition between the attacker and the agent can be modeled as a zero-sum two-player game. Notably, this is not a fair game and is more biased towards the agent for two reasons. First, the sound attack is just single-modal and will not intervene in any visual information obtained by the agent. Second, as will be specified in our methodology, the sound volume of the attacker is bounded via a relative ratio (less than 1) of the sound target. We anticipate constraining the attacker's power and encouraging it to intervene but not defeat the agent with these two considerations. With such a design, we can improve the agent's robustness between the agent and the sound attacker during the game. On the other hand, our environment is more demanding than reality since there are few attackers in our lives. Instead, most behaviors, such as someone walking and chatting past the robot, are not deliberately embarrassing the robot but just a distraction to the robot, exhibiting weaker intervention strength than our adversarial setting. Even so, our experiments reveal that an agent trained in a worst-case setting can perform promisingly when the environment is acoustically clean or contains a natural sound intervenor using a random policy. On the contrary, the agent trained in a clean environment becomes disabled in an acoustically complex environment.

Our training algorithm is built upon the architecture by (Chen et al., 2020), with a novel decision-making branch for the attacker. Training two agents separately (Tampuu et al., 2017) leads to divergence. Hence we propose a joint Actor-Critic (AC) training framework. We define the policies for the attacker based on three types of information: position, sound volume, and sound category. Exciting discoveries from experiments demonstrate that the joint training converges promisingly in contrast to the independent training counterpart.

This work is the first audio-visual navigation method with a sound attacker to the best of our knowledge. To sum up, our contributions are as follows.

- We construct a sound attacker to intervene environment for audio-visual navigation that aims to improve the agent's robustness. In contrast to the environment used by prior experiments (Chen et al., 2020), our setting better simulates the practical case in which there exist other moving intervenor sounds.
- We develop a joint training paradigm for the agent and the attacker. Moreover, we have justified the effectiveness of this paradigm, both theoretically and empirically.
- Experiments on two real-world 3D scenes, Replica (Straub et al., 2019) and Matterport3D (Chang et al., 2017) validate the effectiveness and robustness of the agent trained under our designed environment when transferred to various cases, including the clean environment and the one that contains sound attacker with random policy.

## 2 RELATED WORK

We introduce the research related to our work, including audio-visual navigation, adversarial training in RL, and multi-agent learning.

**Audio-visual embodied navigation.** This task demands a robot equipped with a camera and a microphone to interact with the environment and navigate to the source sound. Existing algorithms towards this task can be divided into two categories according to whether topological maps are constructed or not. For the first category (Gan et al., 2020b; Chen et al., 2021b), LLA (Gan et al., 2020b)

(a)  Audio-Visual Embodied Navigation in Simple Environment.    (b)  Audio-Visual Embodied Navigation in Complex Environment.

Figure 1: **Comparison of audio-visual embodied navigation in clean and complex environment.** (a) Audio-visual embodied navigation in an acoustically clean environment: The agent navigates while only hearing the sound emitted by the source object. (b) Audio-visual navigation in an acoustically complex environment: The agent navigates with the audio-visual input from the source object, with the sound attacker making sounds simultaneously.

plans the robot's navigation strategy based on graph-based shortest path planning. Following LLA, AV-WaN (Chen et al., 2021b) combines dynamically setting waypoints with the shortest path algorithm to solve the long-period audio-visual navigation problem. The second category of methods works in the absence of a topological map (Chen et al., 2021a; 2020). In particular, SAVi (Chen et al., 2021a) aims to solve the audio-visual navigation problem with temporary sound source by introducing the category information of the source sound and pre-training the displacement from the robot to the sound source; AVN (Chen et al., 2020) constructs the first audio-visual embodied navigation simulation platform—SoundSpaces and makes use of Reinforcement Learning (RL) for the training of the navigation policy. As presented in the Introduction, all environments used previously (including SoundSpaces) assume clean sound sources, which is hardly the case in real and noisy life. By contrast, we build an acoustically complex environment that allows a sound attacker to move and change the volume and category of sound at each time step in an episode. In this environment, we train the navigation policy of the agent under the sound attack, which delivers a more robust solution in real applications.

**Adversarial attacks and adversarial training in RL.** In general, adversarial attacks (Tian & Xu, 2021)/training in RL are divided into two classes: learning to attack (Huang et al., 2017; Kos & Song, 2017; Lin et al., 2017; Pattanaik et al., 2018) and learning to defence (Pinto et al., 2017; Gleave et al., 2020; Zhang et al., 2020) that targets on learn a robustness policy by state adversarial through external forces or sensor noise. Our paper falls into the second class. A close work to our method is **S**tate **A**dversarial MDP (SA-MDP) (Zhang et al., 2020) that leverages sensor noise in vision to improve the robustness of the algorithm. The main difference between our method and SA-MDP is that our method precisely permits a sound attacker to move in the room and employs the resulting sound as a distractor, while in SA-MDP, the adversarial state is created arbitrarily and thus not necessarily fits the actual scene. Besides, SA-MDP initially copes with the visual modality; hence, we have changed SA-MDP for the attack of the sound modality for the comparison with our method (See § 4).

**Multi-agent learning.** We design two frameworks similar to independent learning (Tampuu et al., 2017) and multi-agent learning (Sunehag et al., 2018). However, the framework of independent learning does not converge (See § 4). We formulate our learning algorithm as a two-player game employed the property of a centralized critic with decentralized actors (Wang et al., 2021) to guarantee the training convergence of our method in theory (See Theorem 1).

## 3   SOUND ADVERSARIAL AUDIO-VISUAL NAVIGATION

### 3.1   PROBLEM DEFINITION

**Problem modeling of ours.**   We model the agent as playing against an attacker in a two-player Markov game (Simon, 2016). We denote the agent and attacker by superscript $\omega$ and $\nu$, respectively. The game $\mathcal{M} = (\mathcal{S}, (\mathcal{A}^\omega, \mathcal{A}^\nu), \mathcal{P}, (\mathcal{R}^\omega, \mathcal{R}^\nu))$ consists of state set $\mathcal{S}$, action sets $\mathcal{A}^\omega$ and $\mathcal{A}^\nu$, and a joint state transition function $\mathcal{P} : \mathcal{S} \times \mathcal{A}^\omega \times \mathcal{A}^\nu \to \mathcal{S}$. The reward function $\mathcal{R}^\omega : \mathcal{S} \times \mathcal{A}^\omega \times \mathcal{A}^\nu \times \mathcal{S} \to \mathbb{R}$ for agent and $\mathcal{R}^\nu : \mathcal{S} \times \mathcal{A}^\omega \times \mathcal{A}^\nu \times \mathcal{S} \to \mathbb{R}$ for attacker respectively depends on the current state, next state and both the agent's and the attacker's actions. Each player wishes to maximize their discounted sum of rewards, where $R^\omega = r$, $R^\nu = -r$. $r$ is the reward given by the environment at every time step in an episode. Our problem is modeled as following (See Fig. 2(c)):

$$\pi^\star = (\pi^{\star,\omega}, \pi^{\star,\nu}) = \arg\max_{\pi^\omega \in \Pi^\omega}\{\arg\min_{\pi^\nu \in \Pi^\nu}\{G(\pi^\omega, \pi^\nu, r)\}\} \tag{1}$$

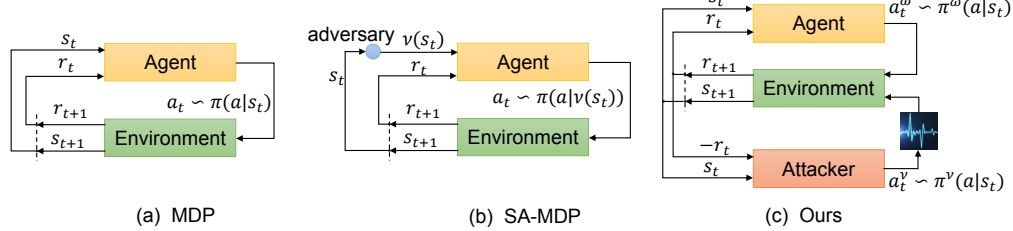

Figure 2: Comparison of different problem modeling methods. AVN is models as an MDP. Our model has an attacker intervention, while the SA-MDP model has an adversary that can map one state in the state space to another state.

Inspired by value decomposition networks (Sunehag et al., 2018) and QMIX (Rashid et al., 2018), we design $G(\pi^\omega, \pi^\nu, r)$ as Equation (2), where $G(\pi^\omega, r)$ and $G(\pi^\nu, r)$ are expected discounted, cumulative rewards of the agent and the attacker respectively.

$$G(\pi^\omega, \pi^\nu, r) = [G(\pi^\omega, r), G(\pi^\nu, r)] \qquad (2)$$

**SA-MDP.** As a reference, we introduce the previous adversarial MDP proposed by (Zhang et al., 2021) as well. In SA-MDP (See Fig. 2 (b)), we optimize $\delta^{adv} := \arg\max_{\delta:\|\delta\|\leq\epsilon} D_{KL}[\pi^\omega(a|s) \mid \pi^\omega(a|s+\delta)]$. Intuitively, the state of SA-MDP is on the ball with radius $\epsilon$.

### 3.2 THEORY ANALYSIS

Why does our model and training mechanism work? Through theoretical analysis, we were pleasantly surprised to find that the observation space of a sound attacker is bounded in the projection space. For more details, see Theorem 1.

**Notations.** $\mathcal{F}, \psi$ stands for short-time Fourier transform and the room impulse response, respectively. $\varsigma_g$ and $\varsigma_\nu$ stand for the waveform signal received by the robot came from a sound source and sound attacker at every time step, respectively. $I_g$ and $I_\nu$ stand for the egocentric visual received by the robot in a clean environment and an acoustically complex environment, respectively. $O$ and $O'$ stand for the observation of what the robot received in a clean environment and an acoustically complex environment, respectively. $O$ and $O'$ is defined as follows: $O = [I_g, \mathcal{F}(\psi_g * \varsigma_g)]$, $O' = [I_\nu, \mathcal{F}(\psi_g * \varsigma_g + \alpha \cdot \psi_\nu * \varsigma_\nu)]$, where $\cdot$ stands for multiply, $*$ stands for time domain convolution, $\alpha$ is the value of action $a^{\nu,\text{vol}}$, $\epsilon$ is a parameter.

Sound attackers do not intervene in the robot's visual modality observation in an acoustically complex environment. So we have :

**Property 1** $I_g = I_\nu$ at every time step in an given episode.

**Assumption 1** The energy of the sound source sent to a robot at every time step for a fixed duration from the same sound set has an upper bound $e$, $\|\mathcal{F}(\varsigma_\nu)\|_2^2 \leq e$, $\|\mathcal{F}(\varsigma_g)\|_2^2 \leq e$.

The distance of intervention is a map function of $O$ and $O'$ : $\Delta : O \times O' \to \mathbb{R}$. We give a distance definition of $O$ and $O'$ as following:

**Definition 1** The distance is : $\Delta(O, O') = \|I_g - I_\nu\|_2^2 + \|\mathcal{F}(\psi_g * \varsigma_g + \alpha \cdot \psi_\nu * \varsigma_\nu) - \mathcal{F}(\psi_g * \varsigma_g)\|_2^2$

Then the sound attacker's observation space $\mathcal{B}_\epsilon(O)$ in an acoustically complex environment based on the above intervention distance is formalized as follows: $\mathcal{B}_\epsilon(O) = \{O' : \Delta(O, O') \leq \epsilon\}$.

**Theorem 1** The observation space of the sound attacker is bounded in the projection space. If a sound attacker's observation is projected into a frequency domain space, the projected result is in a sphere with a radius of $\epsilon$.

**Proof 1**

$$
\begin{aligned}
\Delta(O, O') &= \|I_g - I_\nu\|_2^2 + \|\mathcal{F}(\psi_g * \varsigma_g + \alpha \cdot \psi_\nu * \varsigma_\nu) - \mathcal{F}(\psi_g * \varsigma_g)\|_2^2 && (Def.\ 1) \\
&= \|\mathcal{F}(\psi_g * \varsigma_g) + \mathcal{F}(\alpha \cdot \psi_\nu * \varsigma_\nu) - \mathcal{F}(\psi_g * \varsigma_g)\|_2^2 && (Prop.\ 1, Prop.\ 2) \\
&= \alpha \cdot \|\mathcal{F}(\psi_\nu * \varsigma_\nu)\|_2^2 = \alpha \cdot \|\mathcal{F}(\psi_\nu) \cdot \mathcal{F}(\varsigma_\nu)\|_2^2 && (Prop.\ 2) \quad (3) \\
&= \alpha/2\pi \cdot \|\mathcal{F}(\varsigma_\nu)\|_2^2 && (Prop.\ 2) \\
&\leq \alpha \cdot e/2\pi = \epsilon && (Assump.\ 1)
\end{aligned}
$$

Theorem 1 shows that the attack is bounded and guarantees the rationality of our algorithm. Property 2 in appx. B provides the linear property of the Fourier transform, the convolution theory of the Fourier transform, and the Fourier transform of the Dirac delta function.

### 3.3 Sound Adversarial Audio-Visual Navigation

We propose **S**ound **A**dversarial **A**udio-**V**isual **N**avigation (SAAVN), a novel model for the audio-visual embodied navigation task. Our approach is composed of three main modules (Fig. 3). Given visual and audio inputs, our model 1) encodes these cues and make a decision for the motion of the agent, then 2) encodes these cues and decide how to act for the sound attacker to make an acoustically complex environment, and finally 3) make a judgment for the agent and the attacker and to optimization. The agent and the attacker repeat this process until the agent has been reached and executes the Stop action.

**Environment.** Our work is based on the SoundSpaces (Chen et al., 2020) platform and Habitat simulator (Savva et al., 2019) and with the publicly available datasets: Replica (Straub et al., 2019) and Matterport3D (Chang et al., 2017) and SoundSpaces audio dataset. In SoundSpaces, the sound is created by convolving the selected audio with the corresponding binaural room impulse responses (RIRs) under one of the directions. When a sound attacker emits a chosen sound from its position, the emitted omnidirectional audio is convolved with the corresponding binaural RIR to generate a binaural response from the environment heard by the agent when facing each direction. In this sense, the attacker's sound also considers the reflections on the surface of objects in the environment, making it physically admissible and realistic. The agent's reward is based on how close the robot is away from the goal and whether it succeeds in reaching it. The setting is the same as of the SoundSpaces. The action space of the agent is navigation motions, which is the same as the setting of the SoundSpaces. An environment attacker embodied in the environment must take actions from a hybrid action space $\mathcal{A}^\nu$. For brevity, the abbreviation of superscripts position, volume, and category are set to pos, vol, and cat, respectively. The hybrid action space is the Cartesian product of navigation motions space $\mathcal{A}^{\nu,\text{pos}}$, volume of sound space $\mathcal{A}^{\nu,\text{vol}}$ and category of sound space $\mathcal{A}^{\nu,\text{cat}}$: $\mathcal{A}^\nu = \mathcal{A}^{\nu,\text{pos}} \times \mathcal{A}^{\nu,\text{vol}} \times \mathcal{A}^{\nu,\text{cat}}$. For more details, see appx. C.

**Perception, act, and optimization.** Our model uses acoustic and visual cues in the 3D environment for efficient navigation. Our model has mainly comprised of three parts: the environment attacker, the agent, and the optimizer (See Fig. 3). At every time step $t$, the agent and the attacker receives an observation $O_t = (I_t, B_t)$, where $I$ is the egocentric visual observation consisting of an RGB and a depth image; $B$ is the received binaural audio waveform represented as a two-channel spectrogram. Our model encodes each visual and audio observation with a CNN, respectively, where the output of each CNN are visual vector $f_{I1}(I_t)$ and audio vector $f_{B1}(B_t)$. Then, we concatenate the two vectors to obtain observation embedding representation $e^1 = [f_{I1}(I_t), f_{B1}(B_t)]$. We transform observation embedding representation to calculate state representation by a gated recurrent unit (GRU), $s_t^1 = GRU(e_t^1, h_{t-1}^1)$. An actor-critic network uses $s_t^1$ to predict the action distribution $\pi_\theta^\omega(a_t^\omega | s_t^1, h_{t-1}^1)$ and value of the state $V_\theta^\omega(s_t^1, h_{t-1}^1)$. We also encode visual and audio observation with a CNN for environment attacker, where the output of each CNN are vectors $f_{I2}(I_t)$, $f_{B2}(B_t)$. We then concatenate the two vectors to obtain observation embedding representation $e^2 = [f_{I2}(I_t), f_{B2}(B_t)]$. We also transform observation embedding representation to calculate state representation by a GRU, $s_t^2 = GRU(e_t^2, h_{t-1}^2)$. Three actor-critic networks use $s_t^2$ to predict the action distribution: $\pi_\theta^{\nu,\text{pos}}(a_t^{\nu,\text{pos}} | s_t^2, h_{t-1}^2)$, $\pi_\theta^{\nu,\text{vol}}(a_t^{\nu,\text{vol}} | s_t^2, h_{t-1}^2)$, $\pi_\theta^{\nu,\text{cat}}(a_t^{\nu,\text{cat}} | s_t^2, h_{t-1}^2)$ and value of the state: $V_\theta^{\nu,\text{pos}}(s_t^2, h_{t-1}^2)$, $V_\theta^{\nu,\text{vol}}(s_t^2, h_{t-1}^2)$, $V_\theta^{\nu,\text{cat}}(s_t^2, h_{t-1}^2)$. All actors and critics are modeled by a single linear layer neural network, respectively. Finally, four action samplers sample the next action $a_t^\omega$, $a_t^{\nu,\text{pos}}$, $a_t^{\nu,\text{vol}}$, $a_t^{\nu,\text{cat}}$ from these action distributions generated by AgentActor,

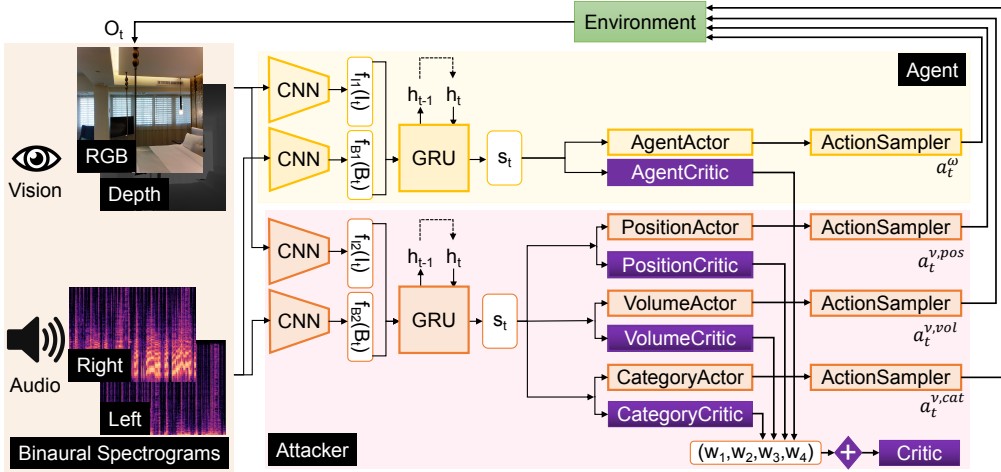

Figure 3: Sound adversarial audio-visual navigation network. The agent and the sound attacker first encode observations and learn state representation $s_t$ respectively. Then, $s_t$ are fed to actor-critic networks, which predict the next action $a_t^\omega$ and $a_t^\nu$. Both the agent and the sound attacker receive their rewards from the environment. The sum of their rewards is zero.

PositionActor, VolumeActor and CategoryActor respectively, determining the agent's next motion in the 3D scene. The total critic is a linear sum of PositionCritic, VolumeCritic, and CategoryCritic (For weights details, see appx. G ). The agent and the environment attacker optimize their expected discounted, cumulative rewards $G(\pi^\omega, r)$ and $G(\pi^\nu, r)$ respectively. The loss of each branch actor-critic network and the total loss of our model as Equation (4).

$$
\begin{aligned}
\mathcal{L}^j &= \sum 0.5 \cdot (\hat{V}_{\theta^j}(s) - V^j(s))^2 - \sum [\hat{A}^j \log(\pi_{\theta^j}(a \mid s)) + \beta \cdot H(\pi_{\theta^j}(a \mid s))] \\
\mathcal{L}^\nu &= 1/3 \cdot (\mathcal{L}^{\nu,\text{cat}} + \mathcal{L}^{\nu,\text{vol}} + \mathcal{L}^{\nu,\text{pos}}) \\
\mathcal{L} &= 1/6 \cdot \mathcal{L}^{\nu,\text{cat}} + 1/6 \cdot \mathcal{L}^{\nu,\text{vol}} + 1/6 \cdot \mathcal{L}^{\nu,\text{pos}} + 1/2 \cdot \mathcal{L}^\omega
\end{aligned}
\tag{4}
$$

where $j \in \{(\nu, \text{cat}), (\nu, \text{vol}), (\nu, \text{pos}), (\omega)\}$. $\hat{V}_{\theta^j}(s)$ is estimated state value of the target network for $j$. $V^j(s) = \max_{a \in \mathbb{A}^j} \mathbb{E}[r_t + \gamma \cdot V^j(s_{t+1}) \mid s_t = s]$. $\hat{A}_t^j = \sum_{i=t}^{T-1} \gamma^{i+2-t} \cdot \delta_i^j$ is the advantage for a given length-T trajectory and $\delta_t^j = r_t + \gamma \cdot V^j(s_{t+1}) - V^j(s_t)$. We optimize the objective follows from *Proximal Policy Optimization* (PPO) (Schulman et al., 2017).

**Algorithms.** Our algorithms are as the following:

---

**Algorithm 1: S**ound **A**dversarial **A**udio-**V**isual **N**avigation

---

**Data:** Environment $\mathcal{E}$, stochastic policies $\omega$ and $\nu$, initial parameters $\theta_0^\omega$ for $\omega$ and $\theta_0^\nu$ for $\nu$, number of updates $N_{\text{iter}}$, $N$.

**Result:** $\theta_{N_{\text{iter}}}^\omega$, $\theta_{N_{\text{iter}}}^\nu$

1 **for** $i=1, 2, ... N_{iter}$ **do**
2     // Run policy $\pi_{\theta_{i-1}^\omega}$ and $\pi_{\theta_{i-1}^\nu}$ in environment for $N$ episodes $T$ time steps ;
3     $\{(o_{t,i}, h_{t-1,i}, a_{t,i}^\omega, a_{t,i}^\nu, r_{t,i}^\omega, r_{t,i}^\nu)\} \leftarrow \text{roll}(\mathcal{E}, \pi_{\theta_{i-1}^\omega}, \pi_{\theta_{i-1}^\nu}, T)$ ;
4     Compute advantage estimates $\hat{A}_1^\omega, \cdots, \hat{A}_T^\omega, \hat{A}_1^\nu, \cdots, \hat{A}_T^\nu$ ;
5     // Optimize $\mathcal{L}(Equation\ 4)$ w.r.t. $\theta^\omega$ and $\theta^\nu$ ;
6     $\theta_i^\omega, \theta_i^\nu \leftarrow$ policy optimize with PPO ;
7 **end**

---

## 4 EXPERIMENTS

**Task.** We use AudioGoal navigation (Chen et al., 2020) tasks to test the acoustically complex environment we designed. In this task, the robot moves in a 3D environment. At each time step $t$, it obtains an RGB and a depth image from its camera, and the binaural microphone receives sensor observations to get $O_t$. As the robot starts navigating, it does not know the scene map; it

must accumulate observations to understand the geometry of the scene during navigation. The robot should use the sound from the audio source to locate successfully and navigate to the target.

**Baselines.** We compare our model to the following baselines and existing works:

1. **Random**: A random policy that uniformly samples one of three actions and executes *Stop* automatically when it reaches the goal (perfect stopping).
2. **AVN (Chen et al., 2020)**: An audiovisual embodied navigation trained in an environment without sound intervention.
3. **SA-MDP (Zhang et al., 2020)**: A model aims to improve the robustness by state adversarial. We adopt its idea but only intervene state of the sound input and do not process the visual information.

Table 1: Comparison of different models in the clean environment under SPL and $R_{\mathrm{mean}}$ metrics. Clean env or cEnv is the abbreviation of clean environment, Acou com env or acEnv is the abbreviation of acoustically complex environment later.

| Method | Clean env. |
|---|---|
| Random | 0.000/-4.7 |
| AVN (Chen et al., 2020) | 0.721/15.1 |
| SA-MDP (Zhang et al., 2020) | 0.590/10.2 |
| SAAVN (ours) | **0.742/16.6** |

Note for Table 2: The row in Table 2 corresponds to how the policy is trained while the column corresponds to how the policy is tested. The environment in the same column of a sub-table is the same. The method in the same row of Table 2 is the same.

Table 2: Comparison of different models in the environment with a specified sound attacker $a^\nu$ under SPL and $R_{\mathrm{mean}}$ metrics.

| Method | Environment with sound attacker $a^\nu$ | | |
|---|---|---|---|
| | $P_{\mathrm{fix}}$ | $P_{\mathrm{random}}$ | P |
| Random | 0.000/-4.5 | 0.000/-4.8 | 0.000/-4.5 |
| AVN (Chen et al., 2020) | 0.676/13.9 | 0.605/11.0 | 0.609/11.0 |
| SA-MDP (Zhang et al., 2020) | 0.456/7.9 | 0.272/5.2 | 0.291/5.4 |
| SAAVN:P(ours) | **0.738/16.6** | **0.659/13.2** | **0.657/13.2** |
| | $V_{\mathrm{fix}}$ | $V_{\mathrm{random}}$ | V |
| Random | 0.000/-4.7 | 0.000/-4.5 | 0.000/-4.5 |
| AVN (Chen et al., 2020) | 0.666/13.5 | 0.535/10.9 | 0.549/11.0 |
| SA-MDP (Zhang et al., 2020) | 0.317/5.2 | 0.202/3.8 | 0.209/3.9 |
| SAAVN:V(ours) | **0.727/16.0** | **0.647/13.6** | **0.648/13.6** |
| | $C_{\mathrm{fix}}$ | $C_{\mathrm{random}}$ | C |
| Random | 0.000/-4.5 | 0.000/-4.7 | 0.000/-4.5 |
| AVN (Chen et al., 2020) | 0.443/9.3 | 0.569/10.4 | 0.576/10.5 |
| SA-MDP (Zhang et al., 2020) | 0.293/4.7 | 0.451/8.0 | 0.461/8.2 |
| SAAVN:C(ours) | **0.666/14.4** | **0.600/12.5** | **0.609/12.7** |
| | $(PVC)_{\mathrm{fix}}$ | $(PVC)_{\mathrm{random}}$ | PVC |
| Random | 0.000/-4.5 | 0.000/-4.7 | 0.000/-4.5 |
| AVN (Chen et al., 2020) | 0.375/9.5 | 0.388/8.2 | 0.389/8.0 |
| SA-MDP (Zhang et al., 2020) | 0.283/4.5 | 0.375/7.2 | 0.368/7.2 |
| SAAVN:PVC(ours) | **0.667/14.9** | **0.557/10.6** | **0.552/10.6** |

**Metrics and symbols.** The evaluations are compared by the following navigation metrics: 1) success weighted by inverse path length (SPL): the standard metric (Anderson et al., 2018) that weighs successes by their adherence to the shortest path; 2) Agent's average episode reward: $R_{\mathrm{mean}}$. The detailed definitions of the symbols used later in this section are as follows: (1) $\pi^{\nu,\mathrm{pos}}$, $\pi^{\nu,\mathrm{vol}}$ and $\pi^{\nu,\mathrm{cat}}$ are abbreviated as P, V and C respectively. (2) $X$, $X_{\mathrm{random}}$ and $X_{\mathrm{fix}}$ denote policies, where $X \in \{P, V, C\}$ is a policy to learn, $X_{\mathrm{random}}$ is a random policy, and when $X \in \{P\}$, $X_{\mathrm{fix}}$ is a determined policy which acts a constant action after random initialization in an episode, while $X \in \{V, C\}$, $X_{\mathrm{fix}}$ is a determined policy which act a constant action set in advance in all episodes. (3) $X, Y_{\mathrm{fix}}$ is abbreviated as $X$, where $X$ is a policy, and $Y \in \{\{P, V, C\} \setminus \{X\}\}$ are set to a determined policy $Y_{\mathrm{fix}}$. (4) SAAVN:X stands for a model variant, where $X$ is a policy for $\pi^\nu$ and $X \in \{P, V, C\}$. (5) SAAVN:X,Y=0.1 is a model variant named SAAVN:X, and $Y_{\mathrm{fix}}$ is set to 0.1, where $X \in \{P, V, C\}, Y \in \{\{P, V, C\} \setminus \{X\}\}$. (6) SAAVN:X,Y=0.1,Z=person6 is a model variant named SAAVN:X, where $Y_{\mathrm{fix}}$ is set to 0.1, $Z_{\mathrm{fix}}$ is set to person6, $X \in \{P, V, C\}, Y \in \{\{P, V, C\} \setminus \{X\}\}, Z \in \{\{P, V, C\} \setminus \{X, Y\}\}$. See appx. H and F for more details.

**Navigation results.** We have built a large number of complex environments with sound attackers therein. The baselines and our SAAVN have been tested several times with different seeds in all these acoustically complex and clean environments to obtain the average navigation ability under each environment. It can be seen from Table 1, 2 that our model achieves the best navigation capabilities in all environments. For more experiments results, see appx. H.

**Trajectory comparisons.** Fig. 4 shows the test episodes for our SAAVN model. The environment of each row in the figure is consistent, and the model of each column is too. SA-MDP failed to complete the task successfully in all two environments. AVN can complete the task in a clean environment but fails in an acoustically complex environment. SAAVN completed the tasks in all two environments. What is more, the navigation track of SAAVN in an environment without intervention is shorter than that of AVN. These fully demonstrate the navigation capabilities of SAAVN. For more trajectory comparisons, see appx. I.

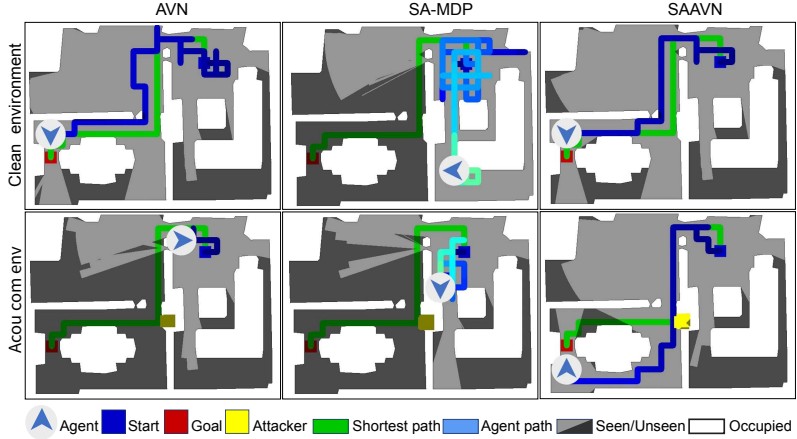

Figure 4: Different models in different environments explore trajectories. The first row in the figure is a clean environment, and the second line is an acoustically complex environment. Acou com env stands for acoustically complex environment.

**Robustness of the model.** In order to verify the robustness of our algorithm, we designed 5 sound attackers, with the settings as follows: $\pi^{\nu,\text{pos}}$ is a learned policy, $\pi^{\nu,\text{cat}}$ is set to person6, while $\pi^{\nu,\text{vol}}$ is fixed to 0.1, 0.3, 0.5, 0.7 and 0.9 respectively. We train AVN, SA-MDP, and SAAVN named SAAVN: P, V=0.1, C=person6 respectively in the same environment created by a designated sound attacker. Then we test them multiple times in the above five environments. Fig. 5 shows the performance of different models under different sound attacks. We found that these sound attackers have certain attack capabilities against these three algorithms. However, it is found that our method's performance decreases more slowly, which fully demonstrates that our method helps to improve the robust performance of the algorithm.

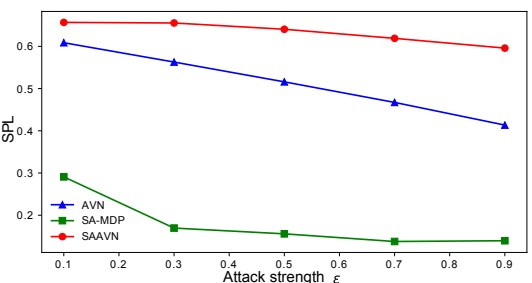

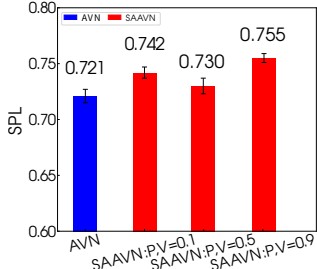

Figure 5: Navigation capabilities under different sound attack strengths.

Figure 6: Performance affect by volume.

**Is the louder the sound attacker's volume, the better?** To verify the navigation ability of an agent that has grown in an acoustically complex environment in a clean environment, we first train our model in the acoustically complex environments where the sound attacker exists and then test the navigation ability of the model by migrating to the environment which the sound attacker is removed. In Fig. 6, SAAVN : P, V=0.1 is short for a model variant named SAAVN : P, V=0.1, C=person6. Others are similar. It can be seen from Fig. 6 that the navigation ability of the attacker is excellent when the volume of the attacker is 0.1 and 0.9, but not very good when the volume is 0.5. The ablation experiments show that simply increasing the sound volume of the attacker does not necessarily lead to better performance. The relationship between the navigation capacity and the volume of the sound attacker is not straightforward and depends on other factors, including the position and sound category. It hence supports the necessity of our method to make the volume policy of the attacker learnable. As such, agents can learn better navigation skills in an acoustically intervened environment.

**Navigation results in the clean environment.** When the sound attacker does not exist, what is the navigation ability of an agent that grows in an acoustically complex environment? The performance of AVN is trained in a simple environment and tested in a clean environment; The performances of SAAVN are trained in acoustically complex environments and tested in a clean environment. It can be seen from Fig. 7 (a) that the ability of an agent that grows up in an acoustically complex

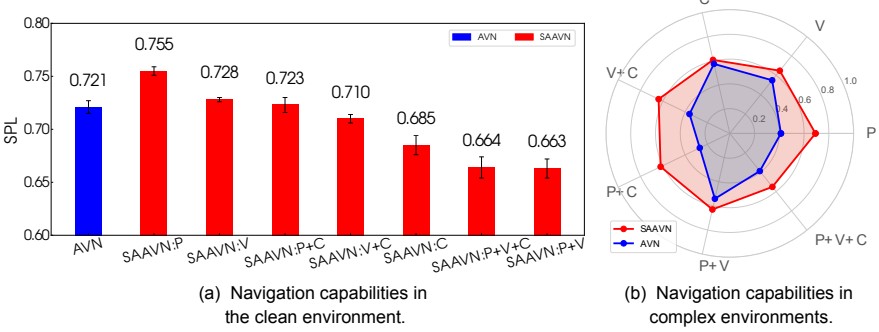

(a) Navigation capabilities in the clean environment.

(b) Navigation capabilities in complex environments.

Figure 7: Navigation capabilities in different environments.

environment to navigate in a clean environment depends on the complexity of the environment. If the environment is acoustically complex, but within a certain range, the ability of the agent will increase; if the environment is too complicated and the changes are relatively great, the navigation ability of the agent is reduced a bit, but not too much. The ablation study shows that the environment should not be too complicated to achieve optimal navigation capabilities.

**Navigation results in acoustically complex environments.** What is the navigation ability of an agent in an acoustically complex environment? In Fig. 7 (b), we can see different sound attackers. As the attack intensity of sound attackers ascends, the navigation capabilities of both AVN (Chen et al., 2020) and ours (SAAVN) decline. However, our method has a relatively small reduction in navigation capability and speed compared with AVN, showing better robustness and navigation capabilities.

**Independent learning framework does not converge in training.** We have also designed a learning framework where the agent and the sound attacker learn independently but not convergent. Our SAAVN employs the multi-agent learning mechanism to train the two policies simultaneously, whose benefit in convergence is empirically demonstrated by Fig. 8.

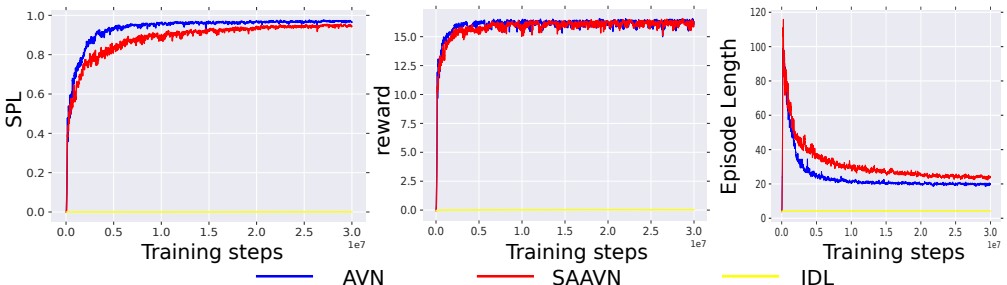

Figure 8: Training curve comparison between AVN, SAAVN, and IDL.

## 5 CONCLUSION

This paper proposes a game where an agent competes with a sound attacker in an acoustical intervention environment. We have designed various games of different complexity levels by changing the attack policy regarding the position, sound volume, and sound category. Interestingly, we find that the policy of an agent trained in acoustically complex environments can still perform promisingly in acoustically simple settings, but not vice versa. This observation necessitates our contribution in bridging the gap between audio-visual navigation research and its real-world applications. A complete set of ablation studies is also carried out to verify the optimal choice of our model design and training algorithm. However, the limitation of our work is that we only assume one sound attacker and have not studied the scenario with two and more attackers. Another limitation of our work is that our current evaluations are conducted in virtual environments. It will make better sense to assess our method on practical cases like a robot navigating in a real house. The above two will be left for future exploration. Since our research is conducted on a simulation platform, it is unlikely to cause a negative social impact in the foreseeable future.

## 6 REPRODUCIBILITY

Our code is available at : appx. L and `https://github.com/yyf17/SAAVN/tree/main`. The algorithm parameters are detailed in appx. G.

Please refer to appx. A for more details about the acoustically clean or simple environment and acoustically complex environment. For more detailed information about the Fourier transform properties, see appx. B.

The dataset Replica and Matterport3D is used in the experiment. For basic information about the dataset, you can refer to appx. L for more details. Please refer to `https://github.com/yyf17/SAAVN/blob/main/dataset.md` for the detailed steps of downloading and processing the dataset.

## 7 ETHICS STATEMENT

The research in this paper does NOT involve any human subject, and our dataset is not related to any issue of privacy and can be used publicly. All authors of this paper follow the ICLR Code of Ethics (https://iclr.cc/public/CodeOfEthics).

## ACKNOWLEDGEMENT

The following projects jointly supported this work: the Sino-German Collaborative Research Project Crossmodal Learning (NSFC 62061136001/DFG TRR169); Beijing Science and Technology Plan Project (No.Z191100008019008); the National Natural Science Foundation of China (No.62006137); Natural Science Project of Scientific Research Plan of Colleges and Universities in Xinjiang (No.XJEDU2021Y003). We gratefully acknowledge the support of MindSpore, CANN(Compute Architecture for Neural Networks) and Ascend AI Processor used for this research. We thank Mingxuan Jing and Zhenhong Jia for their generous help and insightful advice.

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

# APPENDIX

In this section we provide additional details about:

H More experimental results. It mainly includes comparative experiments in a clean environment and the acoustically complex environments on Replica and Matterport3D (As referenced in § 4 of the main paper. ). See Appendix H for more details.

## A    DEFINITIONS.

**Definition 2** *Acoustically clean or simple environment. The acoustically clean or simple environment is described as follows: (1) The number of target sound sources is one. (2) The position of the target sound source is fixed in an episode of a scene. (3) The volume of the target sound source is the same in all episodes of all scenes, and there is no change.*

**Definition 3** *The acoustically complex environment.    The acoustically complex environment referred to in this study is defined as follows: (1) There is one non-target sound source in the scene. (2) The position of the non-target sound source is uncertain, which means that the position of the non-target sound source in the scene is arbitrary. (3) The volume of the non-target sound source is uncertain. It is only known that the maximum volume of the non-target sound is equal to the maximum volume of the target non-target sound source, but the specific volume is uncertain. (4) The sound category of the non-target sound source is uncertain. Only the set to which the non-target sound category belongs is the same as the set to which the target sound source category belongs. (5)The attacker has no physical entity. It is like an invisible ghost. It has no shape, no volume, and no mass. When the agent runs in front of it, it will not block its movement or collide with the agent. This assumption is to simplify the model. These non-target sounds are a major obstacle to audio-visual embodied navigation, which greatly increases the search time.*

**Properties of non-target sound sources** The non-target sound sources have their characteristics: (1) Although non-target and target sounds belong to the same spectrum range, these non-target sounds are not suitable for Gaussian use. Distribution is used for modeling, and noise is suitable for modeling with Gaussian distribution. It is difficult to model both the non-target low voice and the target's natural gas alarm sound. (2) The existence of these non-target sounds is a significant obstacle to audio-visual embodied navigation, which significantly increases the search time. (3) These non-target sounds may not move like the target, or they may explore around the scene. Our work is based on the scope of the above definition.

## B PROPERTIES OF FOURIER TRANSFORM.

**Property 2** *The following three properties of the Fourier transform need to be used when proving the Theorem 1.*
*1. Linearity : The Fourier transform of sum of two or more functions is the sum of the Fourier transforms of the functions. $\mathcal{F}(\boldsymbol{a} + \boldsymbol{b}) = \mathcal{F}(\boldsymbol{a}) + \mathcal{F}(\boldsymbol{b})$. If we multiply a function by a constant, the Fourier transform of the resultant function is multiplied by the same constant. $\mathcal{F}(k \cdot \boldsymbol{a}) = k \cdot \mathcal{F}(\boldsymbol{a})$.*
*2. Convolution : The Fourier transform of a convolution of two functions is the point-wise product of their respective Fourier transforms. $\mathcal{F}(f * g) = \mathcal{F}(f) \cdot \mathcal{F}(g)$.*
*3. Dirac delta function : Fourier transform of Dirac delta function is $\frac{1}{2\pi}$. So $\mathcal{F}(\psi) = \frac{1}{2\pi}$.*

## C ENVIRONMENT.

**SoundSpaces.** SoundSpaces uses the Habitat simulator with the publicly available Replica and Matterport3D environments and the public SoundSpaces audio simulation. The 18 replica environments are grids constructed based on accurate scans of apartments, offices, hotels, and rooms. The 85 Matterport3D environments are real homes and other indoor environments with 3D grids and image scanning. Use SoundSpaces' room impulse response (RIR) to place the audio source and environmental attackers in a 3D environment, and then simulate realistic sound at each position in the scene, where the spatial resolution of the copy is 0.5m, and the spatial resolution of Matterport3D is 1m. The robot can walk in space while receiving real-time egocentric visual and audio observations.

**Rewards.** According to the classic navigation rewards, if the robot successfully reaches the target and executes the *Stop* action, the environment will reward it with $+10$, plus an additional bonus of $0.25$ to reduce the Manhattan distance from the robot to the target. Furthermore, increase the equivalent penalty for this target. Finally, we impose a time penalty of $-0.01$ on each action performed to encourage efficiency.

**Action space.** SoundSpaces maintains a navigability graph of the environment (unknown to the agent). The agent can only move from one node to another if an edge is connecting them and the agent is facing that direction. The action space of the agent are navigation motions: $\mathcal{A}^{\omega} = \{MoveForward, TurnLeft, TurnRight, Stop\}$. An environment attacker embodied in the environment must take actions from a hybrid action space $\mathcal{A}^{\nu}$. The hybrid action space are Cartesian product of navigation motions space $\mathcal{A}^{\nu, \text{position}}$, volume of sound space $\mathcal{A}^{\nu, \text{volume}}$ and category of sound space $\mathcal{A}^{\nu, \text{category}}$: $\mathcal{A}^{\nu} = \mathcal{A}^{\nu, \text{position}} \times \mathcal{A}^{\nu, \text{volume}} \times \mathcal{A}^{\nu, \text{category}}$, where $\mathcal{A}^{\nu, \text{position}} = \mathcal{A}^{\omega}$ is navigation motion space, $\mathcal{A}^{\nu, \text{volume}} = \{0.0, 0.1, 0.2, \ldots, 1.0\}$ is discrete action space with $|\mathcal{A}^{\nu, \text{volume}}| = 11$, $\mathcal{A}^{\nu, \text{category}} = \{'telephone', 'person', \ldots\}$ is discrete action space with $|\mathcal{A}^{\nu, \text{category}}| = 101$ respectively, We experiment with 101 everyday sounds. We use 101 natural sounds without duplication in SoundSpaces, covering various categories: birds, air conditioners, doorbells, door openings, music, computer beeps, fans, talkers, phones, and etc.

## D IMPLEMENTATION DETAILS.

In the following, we provide details of our reinforcement learning (RL) for navigation tasks. The environment attacker embodied in an environment must take actions from an action space $\mathcal{A}^{\nu}$ to make a change of the acoustically simple environment of SoundSpaces. An agent embodied in an environment must take actions from an action space $\mathcal{A}^{\omega}$ to accomplish an end goal.
At every time step, $t = \{0, 1, 2, \ldots, T - 1\}$ the environment is in some state, but the environment attacker and the agent obtain only a partial observation of it in the form of $s_t$. Here T is a maximal

time horizon, corresponding to 500 actions in a scene of a scene for our task. The observation $s_t$ is a combination of the audio, and visual inputs.

The environment attacker develops a hybrid policy $\pi^\nu_{t,\theta} = \pi^{\nu,\text{position}}_{t,\theta} \times \pi^{\nu,\text{volume}}_{t,\theta} \times \pi^{\nu,\text{category}}_{t,\theta}$, where $\pi^{\nu,\text{position}}_{t,\theta} : \mathcal{A}^{\nu,\text{position}} \to \mathbb{R}^3$, $\pi^{\nu,\text{volume}}_{t,\theta} : \mathcal{A}^{\nu,\text{volume}} \to \mathbb{R}^{11}$, $\pi^{\nu,\text{category}}_{t,\theta} : \mathcal{A}^{\nu,\text{category}} \to \mathbb{R}^{101}$ respectively, and where $\pi^\nu_{t,\theta}(a_t \mid s_t, h_{t-1})$ is the probability that the environment attacker chooses to take action $a^\nu \in \mathcal{A}^\nu$ at time t given the current observation $s_t$ and aggregated past states $h_{t-1}$.

Using information about the previous time steps $h_{t-1}$ and current observation $s_t$, the agent develops a policy $\pi^\omega_\theta : \mathcal{A}^\omega \to \mathbb{R}^4$, where $\pi^\omega_{t,\theta}(a_t \mid s_t, h_{t-1})$ is the probability that the agent chooses to take action $a^\omega \in \mathcal{A}^\omega$ at time t given the current observation $s_t$ and aggregated past states $h_{t-1}$.

After the environment attacker and the agent acts, the environment goes into a new state $s_{t+1}$ and the agent and attacker receives individual rewards $r^\omega_t \in \mathbb{R}$ and $r^\nu_t \in \mathbb{R}$ respectively.

The agent optimizes its return, i.e. the expected discounted, cumulative rewards $G^\omega_{\gamma,t} = \sum_{t=0}^{T-1} \gamma^t r^\omega_t$. The environment attacker also optimizes its return, i.e. the expected discounted, cumulative rewards $G^\nu_{\gamma,t} = \sum_{t=0}^{T-1} \gamma^t r^\nu_t$, where $\gamma \in [0,1]$ is the discount factor to modulate the emphasis on recent or long term rewards. The value function $V^\omega_{t,\theta}(s_t, h_{t-1})$ and $V^\nu_{t,\theta}(s_t, h_{t-1})$ is the expected return for agent and attacker respectively. We optimize the particular reinforcement learning objective directly following from Proximal Policy Optimization(PPO). We refer the readers to PPO for additional details on optimization.

We train our model with Adam with a learning rate of $2.5 \times 10^{-4}$. The auditory and visual encoder output are 512 and 512, respectively. We use a one-layer bidirectional GRU with 512 hidden units that take [It, bt] as input. We use an entropy loss on the policy distribution with a coefficient of 0.01. We train the network for $30M$ agent steps on Replica and $60M$ on Matterport3D, which amounts to 150 and 320 GPU hours, respectively.

Following SoundSpace, we first compute the Short-Time Fourier Transform (STFT) with a hop length of 160 samples and a windowed signal length of 512 samples, which corresponds to a physical duration of 12 and 32 milliseconds at a sample rate of 44100Hz (Replica) and 16000Hz (Matterport). STFT gives a $257 \times 257$ and a $257 \times 101$ complex-valued matrix, respectively, for a one-second audio clip; we take its magnitude, downsample both axes by a factor of 4, and take the logarithm. Finally, we stack the left and right audio channel matrices to obtain a $65 \times 65 \times 2$ and a $65 \times 26 \times 2$ tensor.

## E  SOUNDSPACES DATASET IN DETAILS.

We used the SoundSpaces dataset. Table 3 summarizes SoundSpaces dataset,which includes audio renderings for the Replica and Matterport3D datasets. Each episode consists of a tuple: ⟨scene, agent start location, agent start rotation, goal location, audio waveform⟩. Episodes were generated by choosing a scene and a random start and goal location.

Table 3: Summary of SoundSpaces dataset properties

| Dataset | # Scenes | Resolution | Sampling Rate | Avg. # Node | Avg. Area | # Training Episodes | # Test Episodes |
|---|---|---|---|---|---|---|---|
| Replica | 18 | 0.5m | 44100Hz | 97 | 47.24 $m^2$ | 0.1M | 1000 |
| Matterport3D | 85 | 1m | 16000Hz | 243 | 517.34 $m^2$ | 2M | 1000 |

## F  METRICS IN DETAILS.

Next, we narrate the navigation metrics used in Section 4 of the body paper.

1. Success weighted by Path Length (SPL): weighs the successful episodes with the ratio of the shortest path $l_i$ to the executed path $p_i$, SPL $= \frac{1}{N} \sum_{i=1}^{N} S_i \frac{l_i}{\max(p_i, l_i)}$, where $S_i$ is binary indicator of success in episode $i$, and $N$ is the number of episodes.

2. $R_{\text{mean}}$: stands for average episode reward of agent. See Appendix C for more details on reward.

3. Soft Success weighted by Path Length (SSPL, or SoftSPL): Similar to SPL with a relaxed soft-success criteria. $\text{SSPL} = \frac{1}{N} \sum_{i=1}^{N} \max(0, 1 - \frac{d_i^a}{d_i}) \frac{l_i}{\max(p_i, l_i)}$, where $d_i^a$ is the distance from the agent's current position to the goal when episode $i$ is finished, $d_i$ is the distance from the agent's start position to the goal in the episode $i$, $l_i$ is the shortest path, $p_i$ is the executed path.

4. Success Rate (SR): The fraction of completed episodes, the agent reaches the goal within the time limit of 500 steps and selects the stop action precisely at the goal location, $\text{SR} = \frac{1}{N} \sum_{i=1}^{N} S_i$.

5. Average Distance To Goal (DTG): The agent's average distance to the goal when episodes are finished, $\text{DTG} = \frac{1}{N} \sum_{i=1}^{N} d_i^a$, where $d_i^a$ is the distance from the agent's current position to the goal when episode $i$ is finished.

6. Normalized average Distance To Goal (NDTG) : $\text{NDTG} = \frac{1}{N} \sum_{i=1}^{N} \frac{d_i^a}{d_i}$, where $d_i^a$ is the distance from the agent's current position to the goal when episode $i$ is finished, $d_i$ is the distance from the agent's start position to the goal in the episode $i$.

## G  ALGORITHM PARAMETERS.

The parameters used in our model are shown in Table 4.

Table 4: Algorithm parameters

| Parameter | Replica | Matterport3D |
|---|---|---|
| RIR sampling rate | 44100 | 16000 |
| clip param | 0.1 | 0.1 |
| ppo epoch | 4 | 4 |
| num mini batch | 1 | 1 |
| value loss coef | 0.5 | 0.5 |
| entropy coef | 0.02 | 0.02 |
| learning rate | $2.5 \times 10^{-4}$ | $2.5 \times 10^{-4}$ |
| max grad norm | 0.5 | 0.5 |
| num steps | 150 | 150 |
| use gae | True | True |
| use linear learning rate decay | False | False |
| use linear clip decay | False | False |
| $\gamma$ | 0.99 | 0.99 |
| $\tau$ | 0.95 | 0.95 |
| $\beta$ | 0.01 | 0.01 |
| reward window size | 50 | 50 |
| success reward | 10.0 | 10.0 |
| salck reward | -0.01 | -0.01 |
| distance reward scale | 1.0 | 1.0 |
| hidden size | 512 | 512 |
| w1 | 1/6 | 1/6 |
| w2 | 1/6 | 1/6 |
| w3 | 1/6 | 1/6 |
| w4 | 1/2 | 1/2 |

## H  EXPERIMENTS RESULTS IN DETAIL.

It can be concluded from the main paper that AVN's navigation capabilities are better than SA-MDP. So we focus on comparing the navigation capabilities of AVN and our model SAAVN on dataset Matterport3D. We select the model variant SAAVN: PVC for experimental comparison to prove the effectiveness of our model. Later we will show navigation results on dataset Replica.

**Navigation results on dataset Matterport3D.** Table 5 shows the comparative experiments of different models on the dataset Matterport3D in a clean environment. It can be seen from Table 5 that our model is the best under all metrics in a simple or clean environment.

Table 5: Performance comparison of different models, which was tested in a clean environment under all the metrics in detail on dataset Matterport3D. Results are averaged over 5 test runs.

| Method | SPL ($\uparrow$) | SSPL ($\uparrow$) | SR ($\uparrow$) | $R_{mean}$ ($\uparrow$) | DTG ($\downarrow$) | NDTG ($\downarrow$) |
|---|---|---|---|---|---|---|
| Random | 0.000±0.000 | 0.028±0.000 | 0.000±0.000 | -5.0±0.0 | 25.00±0.00 | 1.108±0.000 |
| AVN (Chen et al., 2020) | 0.539±0.002 | 0.558±0.002 | 0.696±0.004 | 18.1±0.2 | 11.85±0.19 | 0.300±0.006 |
| SAAVN:PVC(Ours) | **0.549±0.009** | **0.572±0.008** | **0.698±0.009** | **18.7±0.2** | **11.20±0.11** | **0.282±0.006** |

Table 6: Performance comparison of different models, tested in acoustically complex environments under different metrics in detail on dataset Matterport3D. Results are averaged over 5 test runs. The acoustically complex environment is PVC.

| | SPL ($\uparrow$) | SSPL ($\uparrow$) | SR ($\uparrow$) | $R_{mean}$ ($\uparrow$) | DTG ($\downarrow$) | NDTG ($\downarrow$) |
|---|---|---|---|---|---|---|
| Random | 0.000±0.000 | 0.027±0.000 | 0.000±0.000 | -5.0±0.0 | 25.01±0.00 | 1.116±0.000 |
| AVN (Chen et al., 2020) | 0.397±0.006 | 0.429±0.004 | 0.612±0.005 | 15.3±0.1 | 13.65±0.13 | 0.374±0.005 |
| SAAVN:PVC(Ours) | **0.478±0.006** | **0.508±0.004** | **0.660±0.004** | **17.3±0.1** | **12.11±0.08** | **0.315±0.002** |

Table 7: Comparison of different models in the acoustically complex environments with sound attacker under SPL and $R_{\text{mean}}$ in details on Matterport3D. Results are averaged over 5 test runs.

| Method | Complex env. | |
|---|---|---|
| | $(PVC)_{\text{random}}$ | $PVC$ |
| Random | 0.000/-5.1 | 0.000/-5.0 |
| AVN (Chen et al., 2020) | 0.397/15.3 | 0.397/15.3 |
| SAAVN:PVC(Ours) | **0.473/17.0** | **0.478/17.3** |

The experimental comparison of different models on the dataset Matterport3D in an acoustically complex environment is shown in Table 6. It can be seen from Table 6 that our model is the best under all metrics in all the acoustically complex environments.

Complex env in the Table 7 stands for the acoustically complex environments which includes $(PVC)_{\text{random}}$ and $PVC$. As can be seen from Table 7, for the two acoustically complex environments $(PVC)_{\text{random}}$ and $PVC$, our model wins under both SPL and $R_{\text{mean}}$ metrics.

From the above Table 5, Table 6 and Table 7, it can be concluded that our model achieves better navigation capabilities than AVN in all environments on dataset Matterport3D under all metrics.

**Navigation results on dataset Replica.** The results of comparative experiments in the clean environment of dataset Replica are shown in Table 8. It can be seen from Table 8 that for the dataset Replica, in a simple or clean environment, the performance of our model variants SAAVN:P, V=0.9, C=person6, and SAAVN:P, V=0.1, C=person6 under all metrics are better than AVN's, which fully shows that our model has more robust navigation capabilities in an acoustically simple environment.

The comparative experiments in the acoustically complex environments on the dataset Replica are shown in Table 9. Complex Env in the Table 9 stands for acoustically complex environments. It can be seen from Table 9 that our various model variants achieve better navigation capabilities than AVN respectively in different acoustically complex environments on dataset Replica under all metrics.

The data in Table 8 and Table 9 can be visualized with the histogram of Figure 9. Figure 9 shows the comparative experimental results of dataset Replica in all environments. The two bars on the left of each box in the Fig. 9 are comparisons of the clean environments, and the two bars on the right are comparisons of the acoustically complex environments. It can be seen from the histogram that our model performs better than AVN in both simple clean environment and acoustically complex environment.

Table 8: Comparison of AVN and SAAVN tested in the clean environment under all the metrics in detail on Replica. Results are averaged over 5 test runs.

| Method | SPL ($\uparrow$) | SSPL ($\uparrow$) | SR ($\uparrow$) | $R_{mean}$ ($\uparrow$) | DTG ($\downarrow$) | NDTG ($\downarrow$) |
|---|---|---|---|---|---|---|
| AVN (Chen et al., 2020) | 0.721±0.006 | 0.749±0.005 | 0.897±0.004 | 15.1±0.1 | 0.68±0.06 | 0.059±0.003 |
| SAAVN:P,V=0.1,C=person6(Ours)$^\heartsuit$ | **0.742±0.005** | **0.757±0.006** | **0.960±0.002** | **16.6±0.1** | **0.18±0.04** | **0.017±0.003** |
| SAAVN:P,V=0.5,C=person6(Ours) | 0.730±0.007 | 0.754±0.006 | 0.943±0.004 | 16.4±0.1 | 0.19±0.02 | 0.022±0.003 |
| SAAVN:P,V=0.9,C=person6(Ours)$^\clubsuit$ | 0.755±0.004 | 0.776±0.003 | 0.948±0.007 | 16.4±0.1 | 0.22±0.03 | 0.022±0.003 |
| SAAVN:$P_{fix}$,V,C=person6(Ours) | 0.728±0.002 | 0.742±0.003 | 0.938±0.004 | 15.9±0.1 | 0.38±0.06 | 0.033±0.003 |
| SAAVN:$P_{fix}$,V=0.1,C(Ours) | 0.685±0.009 | 0.700±0.008 | 0.904±0.007 | 15.1±0.1 | 0.68±0.04 | 0.060±0.004 |
| SAAVN:$P_{fix}$,V,C(Ours) | 0.710±0.004 | 0.720±0.005 | 0.941±0.003 | 15.8±0.1 | 0.44±0.07 | 0.038±0.004 |
| SAAVN:P,V=0.1,C(Ours) | 0.723±0.007 | 0.747±0.006 | 0.919±0.004 | 15.6±0.1 | 0.49±0.05 | 0.043±0.003 |
| SAAVN:P,V=0.5,C(Ours) | 0.698±0.005 | 0.705±0.005 | 0.938±0.004 | 15.7±0.1 | 0.57±0.04 | 0.044±0.003 |
| SAAVN:P,V=0.9,C(Ours) | 0.686±0.005 | 0.695±0.004 | 0.954±0.005 | 16.1±0.1 | 0.33±0.06 | 0.025±0.004 |
| SAAVN:P,V,C=person6(Ours) | 0.663±0.009 | 0.692±0.006 | 0.901±0.009 | 15.3±0.2 | 0.52±0.08 | 0.047±0.006 |
| SAAVN:P,V,C(Ours) | 0.664±0.010 | 0.677±0.008 | 0.902±0.012 | 14.8±0.3 | 0.87±0.11 | 0.064±0.008 |

$\heartsuit$:Model variant SAAVN:P,V=0.1,C=person6 ranked second in performance on metrics SPL and SSPL, and ranked first in performance on metrics SR, $R_{mean}$, DTG and NDTG.

$\clubsuit$: Model variant SAAVN:P, V=0.9, C=person6 ranks first in performance on metrics SPL and SSPL, ranked second in performance on metrics $R_{mean}$, DTG and NDTG, and ranked third in performance on the metric SR. Based on the above factors, we believe that model variant SAAVN:P, V=0.1, C=person6 is the best. So in Table 8 and the main paper, we mark model variant SAAVN:P, V=0.1, C=person6 as the best model.

Table 9: Performance comparison of AVN and SAAVN, which is tested in acoustically complex environments with a sound attacker under all the metrics in detail on Replica. Results are averaged over 5 test runs.

| Method | Complex Env. | SPL ($\uparrow$) | SSPL ($\uparrow$) | SR ($\uparrow$) | $R_{mean}$ ($\uparrow$) | DTG ($\downarrow$) | NDTG ($\downarrow$) |
|---|---|---|---|---|---|---|---|
| AVN (Chen et al., 2020) | P,V=0.1,C=person6 | 0.609±0.005 | 0.678±0.004 | 0.712±0.010 | 11.0±0.2 | 2.20±0.06 | 0.194±0.004 |
| SAAVN(Ours) | | **0.657±0.005** | **0.726±0.005** | **0.799±0.006** | **13.2±0.1** | **1.17±0.05** | **0.107±0.005** |
| AVN (Chen et al., 2020) | P,V=0.5,C=person6 | 0.516±0.014 | 0.565±0.011 | 0.712±0.012 | 10.7±0.2 | 2.44±0.09 | 0.207±0.005 |
| SAAVN(Ours) | | **0.689±0.010** | **0.748±0.007** | **0.841±0.009** | **14.4±0.1** | **0.72±0.02** | **0.074±0.001** |
| AVN (Chen et al., 2020) | P,V=0.9,C=person6 | 0.413±0.003 | 0.470±0.004 | 0.657±0.008 | 9.8±0.2 | 2.66±0.10 | 0.228±0.009 |
| SAAVN(Ours) | | **0.692±0.008** | **0.748±0.005** | **0.858±0.009** | **14.7±0.1** | **0.70±0.04** | **0.066±0.003** |
| AVN (Chen et al., 2020) | $P_{fix}$,V,C=person6 | 0.549±0.009 | 0.598±0.007 | 0.723±0.009 | 11.0±0.2 | 2.33±0.06 | 0.196±0.006 |
| SAAVN(Ours) | | **0.648±0.010** | **0.724±0.002** | **0.804±0.016** | **13.6±0.2** | **0.93±0.06** | **0.086±0.005** |
| AVN (Chen et al., 2020) | $P_{fix}$,V=0.1,C | 0.576±0.005 | 0.636±0.008 | 0.694±0.008 | 10.5±0.2 | 2.58±0.10 | 0.222±0.010 |
| SAAVN(Ours) | | **0.609±0.004** | **0.645±0.004** | **0.793±0.004** | **12.7±0.1** | **1.55±0.03** | **0.134±0.002** |
| AVN (Chen et al., 2020) | $P_{fix}$,V,C | 0.361±0.012 | 0.502±0.011 | 0.460±0.010 | 7.6±0.2 | 3.83±0.13 | 0.380±0.013 |
| SAAVN(Ours) | | **0.638±0.010** | **0.683±0.007** | **0.820±0.013** | **13.4±0.2** | **1.14±0.08** | **0.103±0.009** |
| AVN (Chen et al., 2020) | P,V=0.1,C | 0.563±0.005 | 0.630±0.002 | 0.681±0.006 | 10.3±0.1 | 2.60±0.06 | 0.226±0.004 |
| SAAVN(Ours) | | **0.667±0.009** | **0.714±0.008** | **0.826±0.012** | **13.4±0.3** | **1.26±0.10** | **0.102±0.007** |
| AVN (Chen et al., 2020) | P,V=0.5,C | 0.362±0.007 | 0.492±0.006 | 0.469±0.010 | 7.5±0.1 | 3.89±0.04 | 0.384±0.004 |
| SAAVN(Ours) | | **0.644±0.006** | **0.665±0.006** | **0.878±0.003** | **14.4±0.0** | **0.97±0.02** | **0.080±0.001** |
| AVN (Chen et al., 2020) | P,V=0.9,C | 0.269±0.003 | 0.426±0.004 | 0.364±0.006 | 6.2±0.2 | 4.36±0.12 | 0.453±0.012 |
| SAAVN(Ours) | | **0.620±0.004** | **0.673±0.003** | **0.801±0.004** | **13.2±0.1** | **1.29±0.06** | **0.112±0.003** |
| AVN (Chen et al., 2020) | P,V,C=person6 | 0.541±0.004 | 0.590±0.002 | 0.722±0.008 | 11.0±0.1 | 2.30±0.04 | 0.195±0.005 |
| SAAVN(Ours) | | **0.630±0.005** | **0.688±0.004** | **0.777±0.009** | **12.7±0.2** | **1.43±0.08** | **0.129±0.006** |
| AVN (Chen et al., 2020) | P,V,C | 0.389±0.009 | 0.516±0.005 | 0.493±0.012 | 8.0±0.2 | 3.69±0.08 | 0.359±0.008 |
| SAAVN(Ours) | | **0.552±0.004** | **0.598±0.004** | **0.716±0.003** | **10.6±0.1** | **2.56±0.05** | **0.207±0.003** |

## I  TRAJECTORY EXAMPLES IN DETAILS.

**Trajectory comparisons on dataset Replica.** Figure 10 shows the test episodes for our SAAVN model on dataset Replica. The environment of each row in Fig. 10 is consistent, and the model of each column is too. SA-MDP failed to complete the task successfully in all two environments. AVN can complete the job in a clean environment but fail to reach the target in an acoustically complex environment. SAAVN completed the tasks in all two environments. What is more, the navigation track of SAAVN in an acoustically simple environment is shorter than that of AVN. These fully demonstrate the navigation capabilities of SAAVN.

**Trajectory comparisons on dataset Matterport3D.** Figure 11 and Figure 12 shows the test episodes for our SAAVN model on dataset Matterport3D. The environment of each row in the figures is consistent, and the model of each column is too. AVN can complete the task in a clean environment but fails in an acoustically complex environment. SAAVN completed the tasks in all two environments. What is more, the navigation track of SAAVN in an acoustically simple environment is shorter than that of AVN. These fully demonstrate the navigation capabilities of SAAVN.

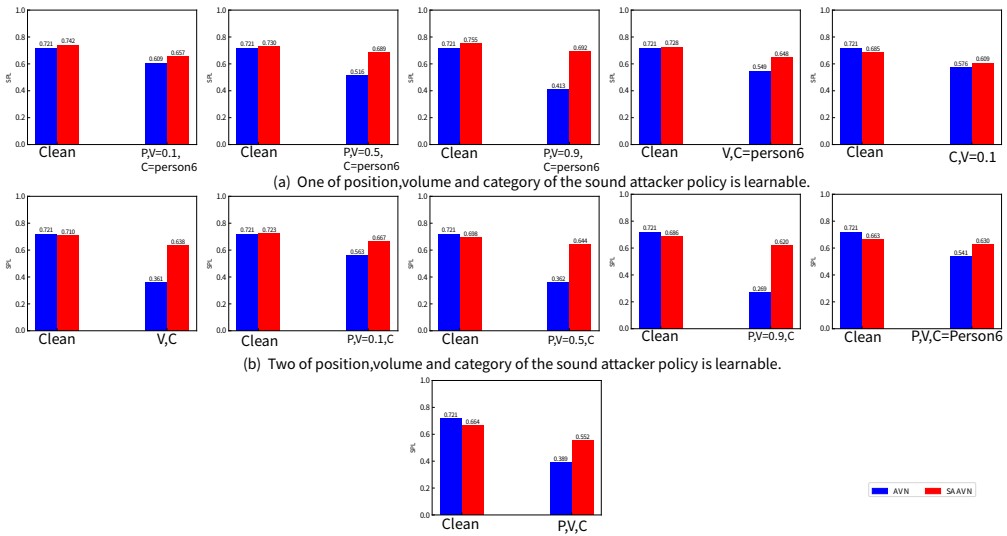

Figure 9: Comparison in the clean environment and different acoustically complex environments between AVN and SAAVN on Replica.

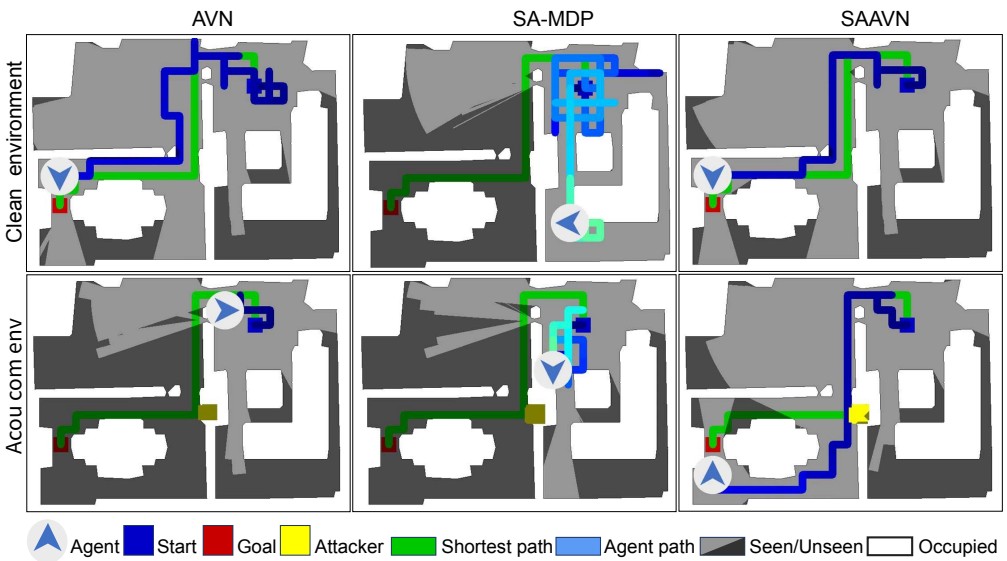

Figure 10: Trajectories of different models in different environments on dataset Replica. The first row in the figure is a clean environment, and the second line is a acoustically complex environment. Each column in the figure represents the same model. Acou com env stands for acoustically complex environment.

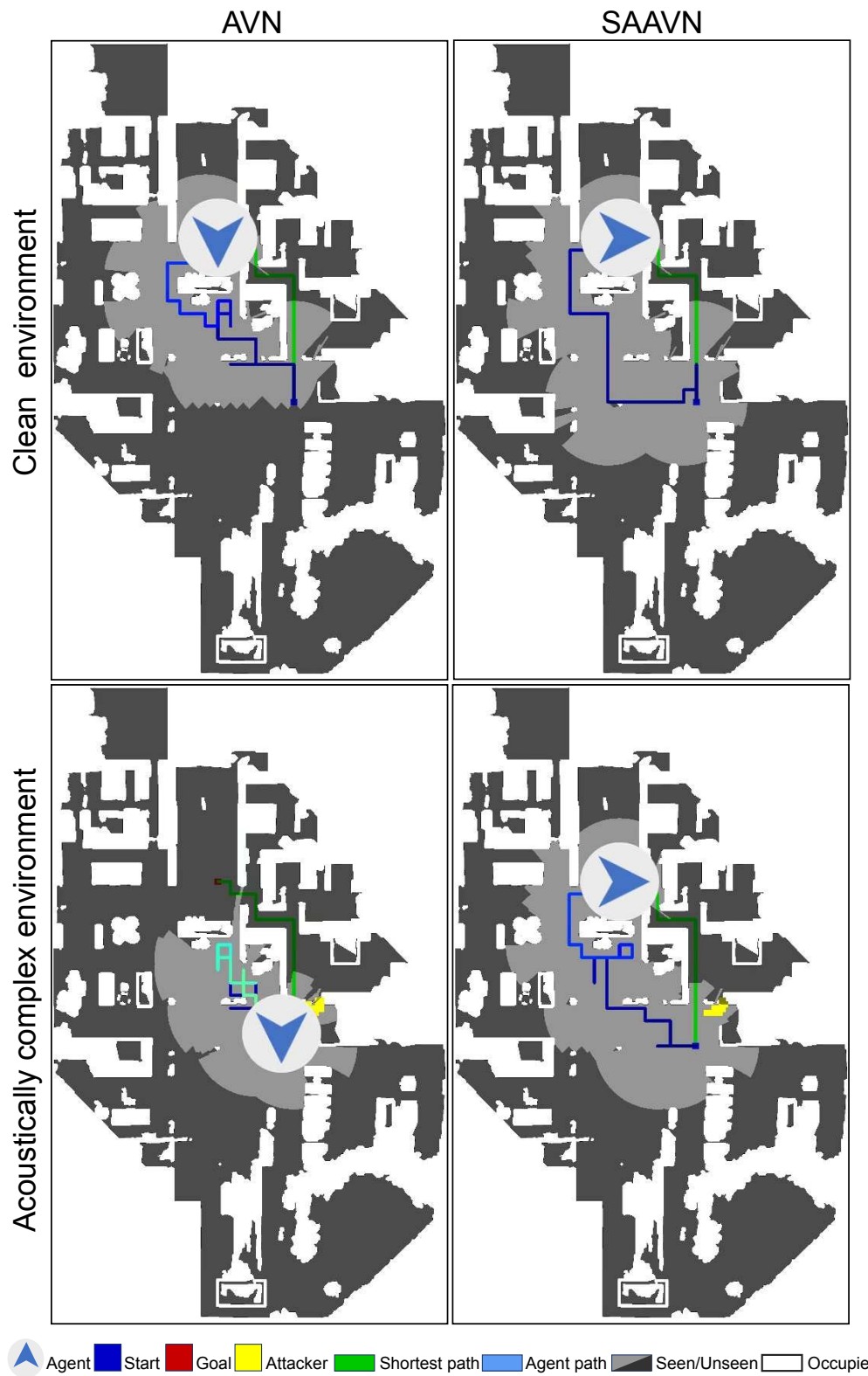

Figure 11: Trajectories of both AVN and SAAVN in different environments on dataset Matterport3D. The first row in the figure is a clean environment, and the second line is a acoustically complex environment. Each column in the figure represents the same model.

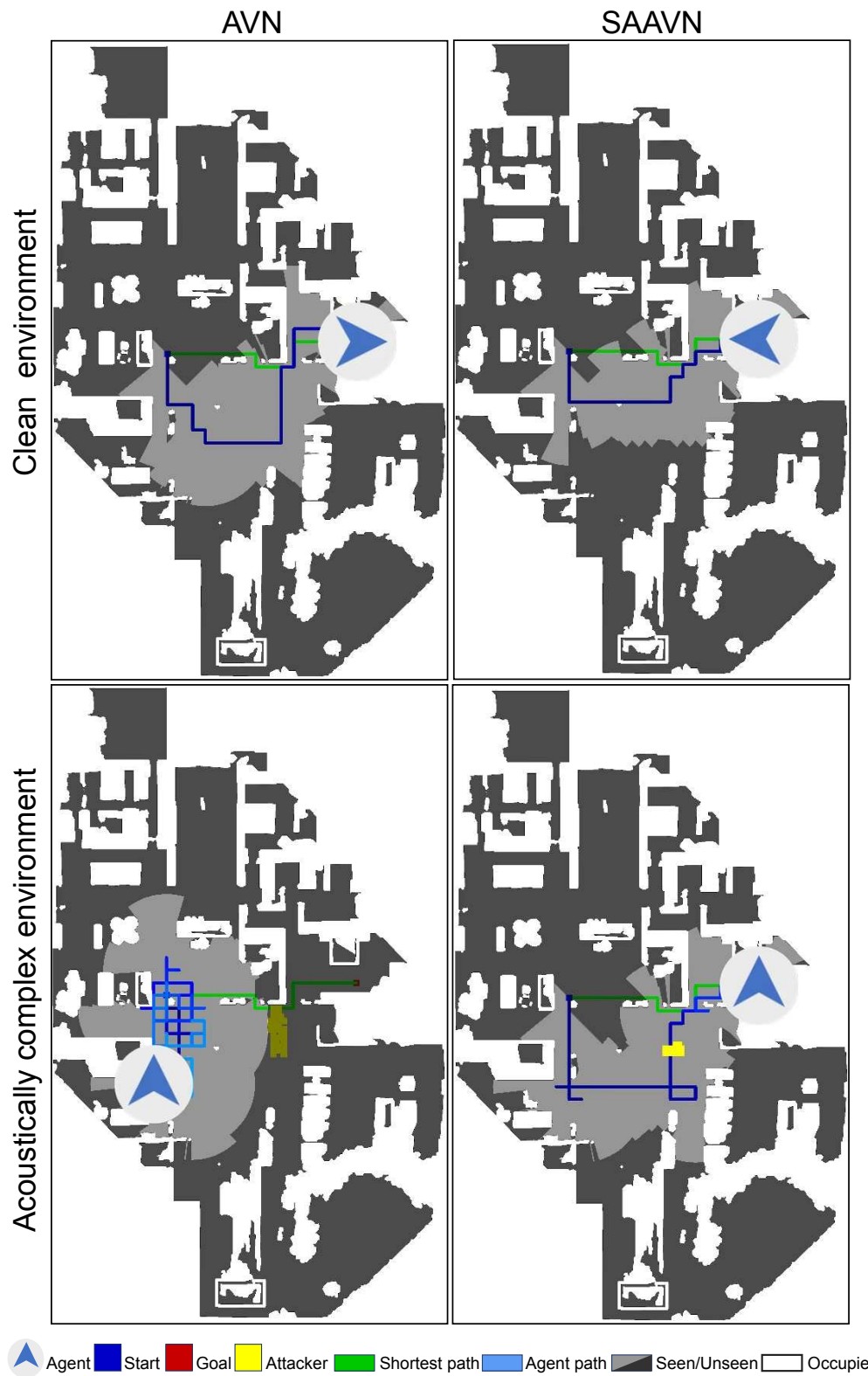

Figure 12: Trajectories of both AVN and SAAVN in different environments on dataset Matterport3D. The first row in the figure is a clean environment, and the second line is a acoustically complex environment. Each column in the figure represents the same model.

## J   INDEPENDENT LEARNING FRAMEWORK DOES NOT CONVERGE IN TRAINING.

Conventional adversarial approaches usually train the agent's policy and the attacker's policy alternately and independently. Instead, our paper employs the multi-agent learning mechanism to train the two policies simultaneously, whose benefit in convergence is empirically demonstrated by Figure 8 (by comparing our SAAVN with IDL). We have also designed a learning framework in the algorithm design process in which the agent and the sound attacker learn independently. It can be seen from Figure 8 that the framework of *independent learning*(IDL) by the agent and the sound attacker is not convergent.

## K   MORE DISCUSSION ABOUT OUR MODEL SAAVN.

Although our work primarily focuses on intervention on the sound modality. We have some discussion on the robustness of SAAVN against the visual modality intervention, the robustness of SAAVN on robot blindness, etc. For more details, please see below.

### K.1   ROBUSTNESS OF SAAVN AGAINST THE VISUAL MODALITY.

Our work primarily focuses on intervention on the sound modality. What is robustness against the visual modality? We add gaussian noise with different std and zero mean to the visual observation (depth image) and evaluation the performance of our model SAAVN on Matterport3D. The result is demonstrated in Table 10 show that When the intensity of intervention increases, performance decreases. However, the decline is slow. This phenomenon shows that our model SAAVN shows better robustness under visual modal intervention.

Table 10: Performance (SPL ($\uparrow$)/$R_{mean}$ ($\uparrow$)) under visual attacking on Matterport3D.

| Method | SPL ($\uparrow$)/$R_{mean}$ ($\uparrow$) |
|---|---|
| w/o noise | 0.478/17.3 |
| std=0.01 | 0.476/17.2 |
| std=0.05 | 0.475/16.9 |

### K.2   ROBUSTNESS OF SAAVN AGAINST BLINDNESS.

When the visual sensor is wholly removed, that is to say, and the robot is blind. What is the performance of our model and AVN? We design and do blindness experiments on Matterport3D. The experimental results in Table 11 show that the performance of SPL and $R_{\mathrm{mean}}$ of model AVN and model SAAVN has decreased. $\Delta SPL$ stands for the deterioration amount under metric SPL for a specific model. The performance value calculates $\Delta SPL$ with removing visual sensor intervention minus the commission value under metric SPL without intervention under metric SPL. $\Delta R_{\mathrm{mean}}$ stands for the deterioration amount under metric $R_{\mathrm{mean}}$ for a specific model. $\Delta R_{\mathrm{mean}}$ is calculated by the performance value with removing visual sensor intervention minus the commission value without intervention for a specific model under metric $R_{\mathrm{mean}}$. The performance degradation of SPL and $R_{\mathrm{mean}}$ of model AVN is greater than that of model SAAVN. It shows that the SAAVN model is more robust than the model AVN when faced with the complete removal of the vision sensor.

Table 11: Performance (SPL ($\uparrow$)/$R_{mean}$ ($\uparrow$)) in the environment with a PVC attacker on Matterport3D. Compared with AVN, our SAAVN performs more robustly without vision, which again exhibits the benefit of our adversarial training.

| Method | Visual + Sound | Sound | $\Delta$ SPL/$\Delta$ $R_{mean}$ |
|---|---|---|---|
| AVN | 0.397/15.3 | 0.196/9.2 | -0.201/-6.1 |
| SAAVN (Ours) | 0.478/17.3 | 0.333/15.1 | **-0.145/-2.2** |

K.3 PERFORMANCE AFFECT BY SLIDING AND SKIPPING MODES OF $a^{\nu,\text{VOL}}$.

The action $a^{\nu,\text{vol}}$ can take by sliding and skipping modes. Suppose the current action $a^{\nu,\text{vol}}$ is 0.5. What is the next step action ? If using the sliding mode, the next action of $a^{\nu,\text{vol}}$ is to select one from neibour actions ([0.4, 0.5, 0.6]) of current action. If using the skip mode, the next action of $a^{\nu,\text{vol}}$ is to select one from whole action space of $a^{\nu,\text{vol}}$ ([0.0, 0.1, 0.2, 0.3, 0.4, 0.5, 0.6, 0.7, 0.8, 0.9, 1.0]). How does the choice of action mode affect the performance of the model SAAVN? The experimental results on Matterport3D in Table 12 show that the performance of model SAAVN adopts the sliding mode is better than that of the skipping mode for action $a^{\nu,\text{vol}}$ under metrics SPL and $R_{\text{mean}}$.

Table 12: Performance under sliding and skipping modes of $a^{\nu,\text{vol}}$ on Replica.

| Method | SPL ($\uparrow$)/$R_{mean}$ ($\uparrow$) |
|---|---|
| Skipping | 0.552/10.6 |
| Sliding | 0.614/13.8 |

K.4 UNSEEN ENVIRONMENTS.

We test our method by assigning the attacker with the sound of a random category that is unseen in training. The following table summarizes the results of AVN and SAAVN. As a reference, we also copy the results from Table 2 that are evaluated under the original "seen" setting. It suggests that SAAVN still performs desirably, particularly for the environments "V" and "C", whereas AVN yields more significant detriment in these two cases (See Table 13, 14, 15).

Table 13: Performance in P and P Unseen environments under SPL ($\uparrow$)/$R_{mean}$ ($\uparrow$) on Replica.

| Method | P | P Unseen | $\Delta$ SPL/$\Delta$ $R_{mean}$ |
|---|---|---|---|
| AVN | 0.609/11.0 | 0.568/10.2 | -0.041/-0.8 |
| SAAVN:P | 0.657/13.2 | 0.612/11.2 | -0.045/-2.0 |

Table 14: Performance in V and V Unseen environments under SPL ($\uparrow$)/$R_{mean}$ ($\uparrow$) on Replica.

| Method | V | V Unseen | $\Delta$ SPL/$\Delta$ $R_{mean}$ |
|---|---|---|---|
| AVN | 0.549/11.0 | 0.445/10.0 | -0.104/-1.0 |
| SAAVN:V | 0.648/13.6 | 0.596/12.3 | -0.052/-1.3 |

Table 15: Performance in C and C Unseen environments under SPL ($\uparrow$)/$R_{mean}$ ($\uparrow$) on Replica.

| Method | C | C Unseen | $\Delta$ SPL/$\Delta$ $R_{mean}$ |
|---|---|---|---|
| AVN | 0.576/10.5 | 0.394/6.7 | -0.182/-3.8 |
| SAAVN:C | 0.609/12.7 | 0.608/13.2 | -0.001/0.5 |

K.5 SAAVN: PVC ACHIEVES THE BEST PERFORMANCE.

Here, we further evaluate different methods in the same attack environment, "PVC," as follows. As expected, SAAVN: PVC achieves the best performance (See Table 16).

K.6 MULTI-MODAL FUSION ABLATION FOR SOUND ATTACKER AUDIO-VISUAL NAVIGATION.

Is concatenation the best choice for multi-modal fusion? We did not explore which choice is the best for multi-modal fusion in the main paper since this is not the main focus of our paper (all methods share the same fusion setting). However, we do think this is an exciting point. Hence, besides concatenation, we have also conducted another fusion strategy using element-wise multiplication. Specifically, we first obtain the visual and audio embedding vectors of the same size (i.e., 512) and then compute the element-wise multiplication between these two vectors. Table 17 reports the performance of SAAVN: PVC in the environment PVC under these two different fusion scenarios.

Table 16: Evaluation of different variants under the same PVC attack environment (SPL ($\uparrow$)/$R_{mean}$ ($\uparrow$)) on Replica.

| Method | Env. | Env. PVC Unseen | $\Delta$ SPL/$\Delta$ $R_{mean}$ |
|---|---|---|---|
| | P | PVC Unseen | |
| SAAVN:P | 0.657/13.2 | 0.286/6.1 | -0.371/-7.1 |
| | V | PVC Unseen | |
| SAAVN:V | 0.648/13.6 | 0.415/9.2 | -0.233/-4.4 |
| | C | PVC Unseen | |
| SAAVN:C | 0.609/12.7 | 0.394/9.0 | -0.215/-3.7 |
| | PVC | PVC Unseen | |
| SAAVN:PVC | 0.552/10.6 | 0.548/10.5 | -0.004/-0.1 |

Interestingly, the new fusion strategy is better than the original concatenation. Upon the observation here, we believe that our proposed method allows further extendibility and can facilitate various following studies.

Table 17: Multi-modal fusion ablation on Replica.

| Fusion | SPL ($\uparrow$) | SSPL ($\uparrow$) | SR ($\uparrow$) | $R_{mean}$ ($\uparrow$) | DTG ($\downarrow$) | NDTG ($\downarrow$) |
|---|---|---|---|---|---|---|
| Concatenation | 0.552±0.004 | 0.598±0.004 | 0.716±0.003 | 10.6±0.1 | 2.56±0.05 | 0.207±0.003 |
| Element-wise multiply | **0.592±0.005** | **0.635±0.005** | **0.768±0.010** | **11.8±0.2** | **2.05±0.09** | **0.164±0.007** |

## L  CODES.

Codes are in the folder named **code** of the supplementary materials. The usage and main introduction of the code are in the readme.MD.

## M  VIDEO DEMONSTRATIONS ON DATASETS REPLICA AND MATTERPORT3D.

Video demonstrations of navigation trajectory in an episode of different models in the clean environment and acoustically complex environments are in the folder named **demoVideo** of the supplementary materials. There are two subfolders under this folder named DemonstrationOnDatasetReplica and DemonstrationOnDatasetMatterport3D, used to store trajectory videos on the dataset Replica the dataset Matterport3D, respectively. Each MP4 file is named according to the format: xxx_in_yyy_env_zzz, where xxx represents the name of the model, $xxx \in \{AVN, SAAVN\}$, yyy represents the type of environment, $yyy \in \{simple, complex\}$, and zzz represents the name of the datasets, $zzz \in \{Replica, Matterport3D\}$.

## N  BROADER IMPACT.

Our research will help service robots provide better navigation services in home life scenes and indoor office scenes. The setting of our research work is based on the complex acoustic environment. The gap between this research setting and the actual application setting is small, conducive to applying the research results to real application scenarios. The limitation of our work is that we only assume one sound attacker and have not studied the scenario with two and more attackers, which will be left for future exploration.

