# OpenReview forum: "Sound Adversarial Audio-Visual Navigation"
_ICLR.cc/2022/Conference — ICLR 2022 Poster_

### Official Review · Reviewer_i5Vv · 2021-11-01

**Correctness:** 4
**Technical Novelty And Significance:** 3
**Empirical Novelty And Significance:** 3
**Recommendation:** 6
**Confidence:** 3

**Main Review:**

The main contribution is the addition of adversarial attacker in training AVN. This can be seen as a complex type of noise augmentation. The results are generally positive, showing significant improvements over baselines.

Ablation studies w.r.t different test environments (e.g. different positions, volumes, noise types) are reasonable. However, it would be good to see ablations w.r.t. design choices made. For example, if the noise sources are random P,V,N instead of adversarial, what would the performance be? Is concatenation the best choice for multi-modal fusion?

The paper needs to be proofread again – singular-plural errors (e.g. space are …), grammatical errors (e.g. why are our model work?), spelling (e.g. neibour)


**Summary Of The Paper:**

The paper proposes an interference-robust training method for audio-visual navigation. Unlike existing approaches that focus on clean environments, the system is trained in simulated acoustically complex environments. A single-source adversarial attacker is introduced, which determines position, noise type and volume that would make the agent to suffer most. This is claimed to improve robustness to random attacks in AVN. Results are compared to a small selection of previous works.

**Summary Of The Review:**

The paper presents interesting solution for AVN in noisy conditions. The results are better than baselines, but the paper would benefit from more ablations.

---

### Official Review · Reviewer_TLMn · 2021-11-02

**Correctness:** 4
**Technical Novelty And Significance:** 3
**Empirical Novelty And Significance:** 4
**Recommendation:** 8
**Confidence:** 4

**Main Review:**

Main questions
1. In table 2, the SAAVN:PVC is the final proposed method, but it is not the best performing one compared to P, V, C. Why?
2. The author mentioned "SA-MDP: A model aims to improve the robustness by state adversarial. We adopt its idea but only intervene state of the sound input and not process the visual input". Then, SA-MDP should be better than AVN. Why does this model keep worse than AVN? If the experimented environment includes visual attack as well as audio attack, could the SA-MDP  be the best?
3. Is this work the first audio-visual navigation method with attackers?

Minors: some notations should be 'italic' font in sentences. For example, in the sentences between eq (1) and eq (2), 'G' should be 'italic' font if they denoting 'G' in eq (1) and (2). In other parts, there seems similar typos.

**Summary Of The Paper:**

This paper addressed the audio-visual navigation task in the environment of sound attacker. The authors formulated the problem as zero-sum game between sound attacker and the agent, and provide a reasonable proof to their formulation and solution. They showed better results than existing works which don't considered this kind of attack (but, I am not sure for SA-MDP).

**Summary Of The Review:**

The environment with the attacker is practical and interesting. The authors well-formulated the action of the agent and attacker.
Also, including the appendix, they tried to provide various experiments and analyses.
BTW, I have some questions to convince the efficacy of the proposed method.

---

> ### Author Response · Authors · 2021-11-25
> **Thank you for increasing the score!**
>
> Dear Reviewer,
>
> We notice that you have increased the score. Thank you very much! Our paper won't be better without your nice suggestions.
>
> Thanks again.

---

### Official Review · Reviewer_yUmi · 2021-11-03

**Correctness:** 3
**Technical Novelty And Significance:** 3
**Empirical Novelty And Significance:** 3
**Recommendation:** 8
**Confidence:** 4

**Main Review:**

Strengths:

+ The motivation is clear and the proposed sound adversarial audio-visual navigation is interesting to me. Beyond clean environments with single sound sources, this work explores more complex environments considering potential attacks from non-target sounds in audio-visual navigation.

+ A joint training paradigm for the agent and the attacker is proposed.

+ Experimental results on Replica and Matterport3D can validate the superiority of the proposed method over recent approaches.

Weaknesses:

- Some important details are missing.
(1) How does visual information help AudioGoal navigation? The perceived visual content is mostly unrelated to target sound sources. So, what cues we can obtain from the RGB frames and depth maps to support navigating to correct sound sources?
(2) Whether the agent could be stuck in certain false local states due to the attacks? If so, how does the proposed joint learning method help to mitigate the problem?

- Fairness of evaluation. It seems that the attacks are different when attacking different methods during evaluation. Did the authors try to add the same attacks during measuring different approaches?

- Unseen Environments. To measure robustness, an important study is to investigate whether the trained model can perform well in unseen environments. Did the authors let the agents explore unseen environments (e.g., containing attacks from unseen sounds)?




**Summary Of The Paper:**

The authors address the audio-visual navigation task in this paper. Previous works usually assume that the interacted environment is clean containing solely the target sound. Different from them, this work explores an acoustically complex environment in which, besides the target sound, there exists a sound attacker playing a zero-sum game with the agent. Specifically, the attacker can move and change the volume and sound category to fool the agent while the agent tries to defend against the attack and navigate to the goal under the intervention. Experiments on Replica and Matterport3D can verify the effectiveness and the robustness of proposed model.

**Summary Of The Review:**

Overall, this is an interesting paper to explore audio-visual navigation in more complex environments and the proposed sound adversarial audio-visual navigation is well-motivated. The extensive experiments can demonstrate the superiority of the proposed method in handling sound attacks. But, I do have some concerns about the method and results. My current rating is borderline accept. But, I would like to upgrade my rating if the authors can address my concerns during rebuttal.

***Post-Rebuttal***

The authors successfully addressed my concerns in the rebuttal. I am happy to upgrade my rating.

There is a very recent work on audio-visual adversarial robustness. I'd like to suggest the authors to discuss this relevant work in the revised paper.

[1] Tian, Yapeng, and Chenliang Xu. "Can audio-visual integration strengthen robustness under multimodal attacks?." Proceedings of the IEEE/CVF Conference on Computer Vision and Pattern Recognition. 2021.

---

> ### Author Response · Authors · 2021-11-24
> **Thank you for increasing the score!**
>
> Dear Reviewer,
>
> We notice that you have increased the score. Thank you very much! Our paper won't be better without your nice suggestions. We are happy to cite and discuss the mentioned related work in our revised paper (once we can edit the paper).
>
> Thanks again.

---

### Decision · Program_Chairs · 2022-01-20

**Decision:**

Accept (Poster)

**Comment:**

This paper addresses audio-visual navigation tasks where a reinforcement learning agent perceives visual RGB and binaural audio inputs, rendered in a first-person perspective 3D environment, and is tasked to navigate to the audio source. The authors propose to make the RL navigation policy robust, by training the agent with additional adversarial audio perturbations. These perturbations consist of an adversarial "ghost" agent (attacker) that emits noise perturbations volume, position and category determined by policies that are trained to maximise the negative rewards for the navigation agent in zero-sum game. The agent is then evaluated on the simulated Replica and MatterPort3D environments and compared to a few baselines. The authors conduct a large number of ablation experiments.

The three reviewers were globally positive about the paper, regarding the motivation, joint training of the agent and attacker, and experimental evaluation. Reviewer TLMn had questions about specific results and ablations of existing baselines, whereas reviewer i5Vv had questions about random noise ablations - the authors provided responses for these questions. Outstanding requests were about proofreading.

After rebuttal and discussion, the scores for this paper are 6, 8 and a weak 8 (or 7), i.e., an average of 7, and thus I believe that the paper meets the conference acceptance bar.